# Intralocus sexual conflict can resolve the male-female health-survival paradox

C. Ruth Archer[1], Mario Recker [ID] [1,2], Eoin Duffy[1] & David J. Hosken[1]

At any given age, men are more likely to die than women, but women have poorer health at older ages. This is referred to as the "male-female, health-survival paradox", which is not fully understood. Here, we provide a general solution to the paradox that relies on intralocus sexual conflict, where alleles segregating in the population have late-acting positive effects on male fitness, but negative effects on female health. Using an evolutionary modelling framework, we show that male-benefit, female-detriment alleles can spread if they are expressed after female reproduction stops. We provide support for our conflict based solution using experimental *Drosophila* data. Our results show that selecting for increased late-life male reproductive effort can increase male fitness but have a detrimental effect on female fitness. Furthermore, we show that late-life male fertility is negatively genetically correlated with female health. Our study suggests that intralocus sexual conflict could resolve the health-survival paradox.

---

[1] Science and Engineering Research Support Facility, University of Exeter, Penryn Campus, Penryn, Cornwall TR10 9FE, UK. [2] College of Engineering, Mathematics and Physical Sciences, University of Exeter, Penryn Campus, Penryn, Cornwall TR10 9FE, UK. Correspondence and requests for materials should be addressed to D.J.H. (email: D.J.Hosken@exeter.ac.uk)

While we broadly understand why mortality risk rises as fertility and general performance decline with age[1], it is less clear why the tempo and severity of these changes often differ between the sexes. In humans, survival, fertility, and performance show sex-specific patterns of decline with age. Strikingly, women stop reproducing decades before dying, while men can reproduce throughout their adult lives. Additionally, men are more likely to die than women in most age-classes, but are healthier than women late-in-life[2,3]. To be clear, this is not just due to the selective loss of low quality males, as female mortality rates are lower than male rates at nearly all ages[2] despite poorer female health. This sex difference has been termed the "male–female, health-survival paradox", and while its causes are not well understood, some resolution of it is needed if we are to ensure healthy aging as human lifespan increases[4].

The sex-specific mortality element of the health-survival paradox is widespread as women outlive men in most countries studied[5]. There is also abundant evidence that older men are healthier than older women in many countries, as although women live longer than men, they experience a smaller proportion of their lives in good health[6]. Men may even experience more time in good health than women, despite dying earlier[7]. It is important to note however that the magnitude, and even direction, of sex differences in physical performance relies on how performance is measured[5]. That is, women may have a lower chance of developing some diseases (e.g. heart disease) but are more likely to suffer from chronic, non-life-threatening conditions[8]. Social and behavioural differences might contribute to these sex differences in survival and health. For example, men are often more likely to engage in risky behaviours (e.g. smoking) than women, but sex differences in mortality are present even in communities where both sexes avoid risky behaviours[9]. Similarly, men and women with similar medical conditions are equally likely to report health problems[10]. While sex differences in behaviour may contribute towards the male–female, health-survival paradox, the consistent observation of lower female mortality but poorer female health at older ages in human populations across the world suggests that this paradox has, at least in part, a genetic basis[2]. To date, most biological mechanisms implicated in the paradox focus on understanding why men die earlier than women (reviewed in refs. [2,11]) rather than on why late-life health differs between the sexes. Here we focus on the health aspect of the paradox and suggest that intralocus sexual conflict might explain why women are less healthy than men late-in-life.

Intralocus sexual conflict occurs when the sexes have different optimal values for a shared trait with a common genetic basis. For example, male broad-horned flour beetles develop enlarged mandibles and males with larger mandibles have higher fitness. However, daughters of males with large mandibles have lower fitness because of the masculinisation of the body that occurs with these genotypes[12]. This means that alleles associated with mandibles are subjected to an intersexual tug-of-war over optimal values, with high fitness male genotypes making low fitness females. This type of conflict means that the alleles encoding a high-quality male often produce low quality females and vice versa[12]. A role for sexual conflict in aging, lifespan and sex differences in health has been recognised for years[13,14]. However, to the best of our knowledge, sexual conflict has not been considered as a driver of the male–female, health-survival paradox. Given the existence of the menopause, which enables selection to bias allelic values towards male-benefit late-in-life, there is enormous potential for sexual conflict to be at the heart of this paradox and by recognising its role, we may better understand what (if anything) we can do about it. The aim of this paper is simply to highlight that as well as explaining sex differences in health and aging in general, intralocus sexual conflict could be central to a long-standing puzzle in medical sciences: why do men die, while women suffer?

Intralocus sexual conflict could cause sex differences in health if there are sexually antagonistic alleles segregating in a population that increase male fitness but reduce female health. Fitness and health are different concepts. Fitness can be thought of as the number of offspring an individual contributes to the next generation[15], and is often measured as lifetime reproductive success, while health reflects how well individuals maintain homoeostasis. Typically, healthier individuals are fitter. For example, more attractive people are likely to have high fitness (because they can attract more mates) and are also less likely to report ill health[16]. However, health and fitness are at least partially decoupled in humans because after the menopause female fitness plummets while health does not to anywhere near the same degree (after menopause direct fitness contributions fall to zero, but health declines are not as severe; not all females experience a complete loss of homoeostasis). Furthermore, sexually antagonistic alleles exist in human populations. For example, men that look particularly masculine tend to have more short-term partners, which is likely to increase their fitness[17], and are more resistant to respiratory conditions[18]. On the other hand, a masculine appearance in women is associated with a greater risk of developing these conditions[18]. Clearly, females that express alleles that are positively associated with male fitness can experience reduced health.

For intralocus sexual conflict to explain the health-survival paradox, male-benefit sexually antagonistic alleles with late-acting effects must accumulate. This is entirely feasible because women experience the menopause. This means that selection against any alleles with costly effects when expressed in females will weaken dramatically once women undergo the menopause and stop reproducing, because these alleles can only have indirect effects on female fitness. However, in men there will be selection for male-benefit alleles over the entire lifespan because men can keep reproducing until advanced ages[19]. This would allow late-acting, male-benefit sexually antagonistic alleles to spread and accumulate in the human genome and reduce female health late-in-life, as females carrying late-acting male-benefit alleles express trait values closer to male than female optima.

To formally test this hypothesis, we assessed whether a male-benefit, sexually antagonistic allele could spread through a diploid population using an evolutionary modelling framework. We show theoretically that under biologically realistic assumptions of costs and benefits, such antagonistic alleles can accumulate. Using *Drosophila* model systems, we then assessed whether sexual conflict solutions are feasible by testing whether populations evolving with selection for late-life male reproduction, but with no direct selection on females (as is the case for post-menopausal women), developed late-life costs to females. Finally, we tested whether late-life male fertility and female performance are negatively associated across a range of standardised genotypes (iso-female lines). Our data broadly support the predictions and suggest that intralocus sexual conflict could help explain the male–female, health-survival paradox.

## Results

**The model.** We devised a population genetic model to track the evolutionary dynamics of a sexually antagonistic allele through a diploid population (Methods). Simulating this model under a wide range of assumptions regarding male-fitness benefits and female fitness costs (Fig. 1), we found that if a sexually antagonistic allele improves male fitness and reduces female health without affecting female fitness, then there is no selection against

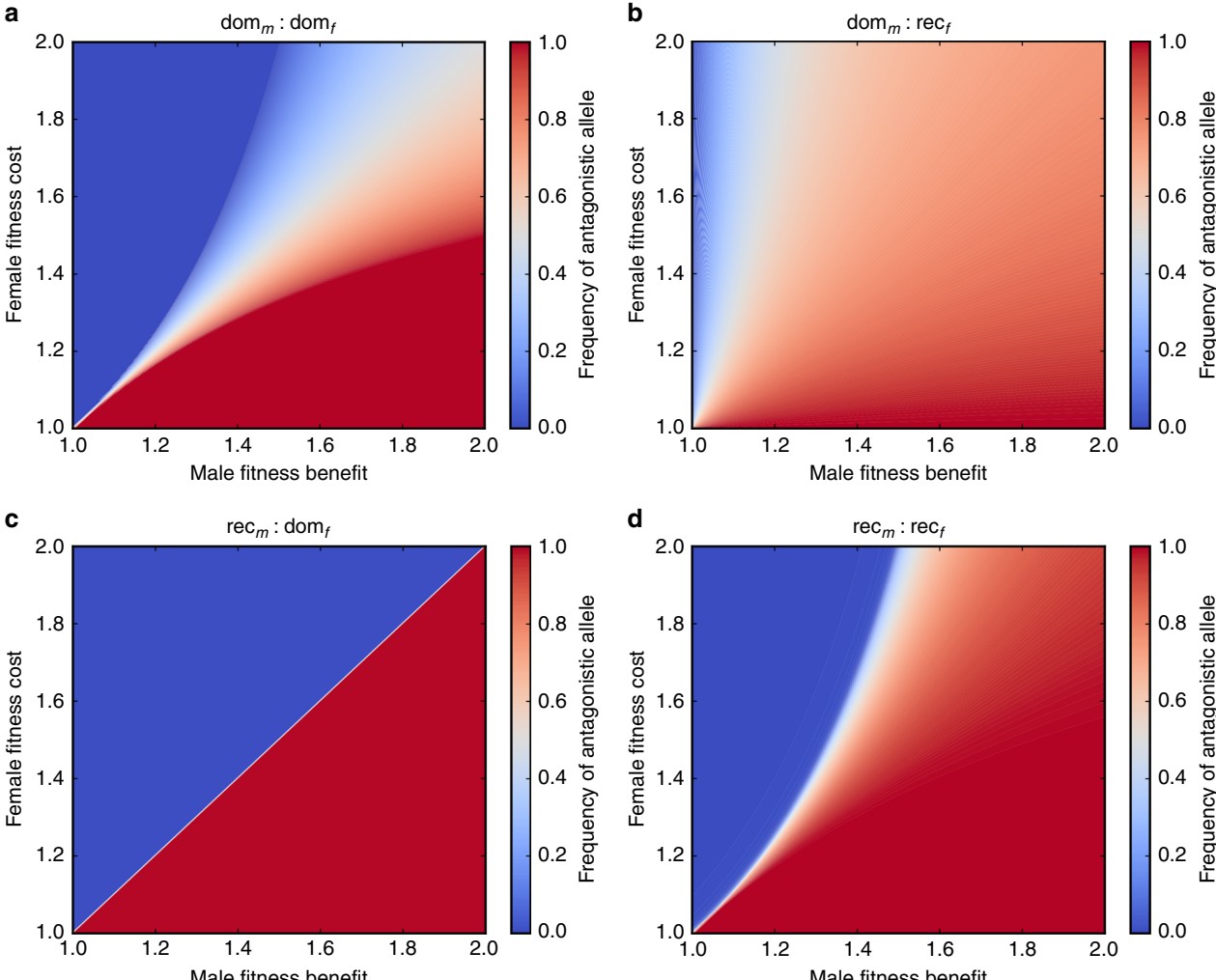

**Fig. 1** Population-level allele frequencies are determined by sex-specific fitness costs and benefits. The graphs show equilibrium allele frequency from simulating our two-sex models over a wide range of male-fitness benefits (relative reproductive success of males with the allele compared to males lacking the allele) and female-fitness costs (relative reproductive success in females with the allele compared to females lacking the allele). The red regions show the conditions where the sexually antagonistic allele goes to fixation, the blue regions indicate where the allele will be lost and the white/colour mixed regions show where both allelic variants coexist. Each panel assumes a different mode of inheritance: the allele is dominant in both sexes (**a**), dominant in males but recessive in females (**b**), recessive in males but dominant in females (**c**) or recessive in both sexes (**d**)

it and it will spread throughout the population. However, a male-benefit allele could affect female fitness by reducing female health, even if it acted after the menopause. This is because older women can care for grandchildren and gain indirect fitness by improving their grandchildren's survival[20]. If a male-benefit allele reduces female health so severely that it impairs care, it may reduce female fitness. In this case, the spread of a male-benefit allele depends on how much that allele improves male fitness, relative to how much it reduces female fitness. This is clearly shown in Fig. 1a when the fitness costs of females expressing the allele are low, then the allele is likely to spread even if males only receive a modest fitness benefit. As the costs to females rise and the male benefits fall, then the allele will become less prevalent in the population and it will eventually be lost. The balance of costs and benefits that favour the spread of a sexually antagonistic allele depend on the nature of that allele, i.e. whether it is dominant (Fig. 1a) or recessive (Fig. 1d) in both sexes, or has sex-specific patterns of expression (Fig. 1b, c). However, the most important thing to note is that alleles reducing post-menopausal female health exist over a broad range of parameter space regardless of the genetic detail, and under some conditions non-antagonistic

alleles are likely to be rare. This strongly suggests that late-acting male-benefit, female-cost alleles are likely to be common and could therefore be responsible for the relatively unhealthy aging of females.

**Selecting on late-life male fertility affects female lifespan.** Using an insect model (*Drosophila simulans*), we then tested whether selecting for increased late-life male reproductive success had a negative impact on females. After 12 generations of selection, we found there was a negative relationship between the fitness ranks of the sexes, such that that in populations where there was the greatest increase in late-life male fertility, there was the greatest reduction in female longevity (correlation coefficient = −0.929; $t = -4.38$, df = 3, $P = 0.022$) (Fig. 2).

**Association between late-life male fitness and female health.** We then used another insect model (*D. melanogaster*) to directly assess female health associations with late-life male fitness. Across isolines (standardised genotypes), there was a negative correlation between male fertility late-in-life and a measure of physiological

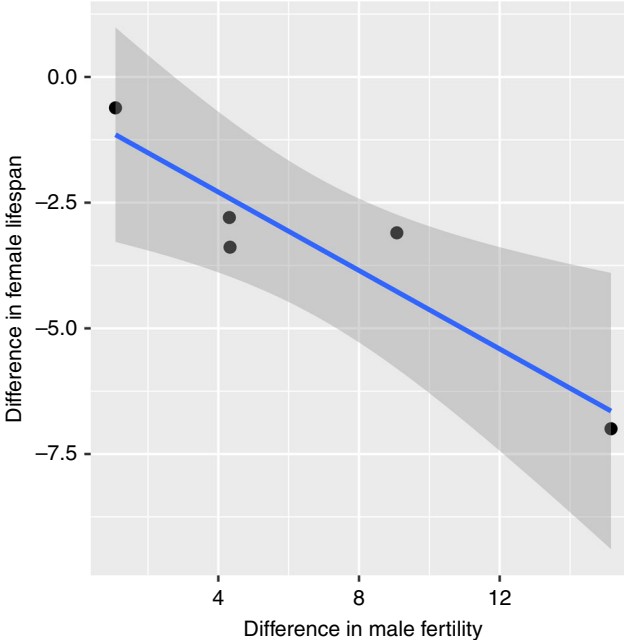

**Fig. 2** Detrimental effect of selection for late-life male fertility on female longevity (=net health). Changes in female lifespan were regressed against improvements in male fertility in populations subject to artificial selection for male late-life fertility relative to the stock population from which experimental flies were derived (correlation coefficient = −0.93; $P$ = 0.022). Thus the stock population acts as a baseline against which evolution was assessed. If a value is equal to zero, the trait average in the selected population is identical to the trait average in the control population. As the value increases, the experimental population has an increasingly higher trait value relative to the control population. As the value declines, the experimental population has a lower trait value than the control. Line averages and 95% confidence intervals are shown

health (the vertical distance females climbed in geotaxis assays) (correlation coefficient = −0.715; $t$ = −2.51, df = 6, $P$ = 0.046). That is, for genotypes where males had large numbers of offspring late-in-life, females performed particularly badly in negative geotaxis assays (one measure of physiological health) (Fig. 3a). However, there was no relationship between male fertility late-in-life and a second measure of female health, recovery time from anaesthesia (correlation coefficient = 0.257; $t$ = 0.596, df = 5, $P$ = 0.577) (Fig. 3b).

## Discussion

Human females tend to live longer but are in poorer health than males late-in-life. This is the health-survival paradox, and given that females have higher survival at any given age despite being in poorer health, this cannot be solely due to selective disappearance of low quality males. We propose a very general resolution to this paradox—intralocus sexual conflict—and both theoretical and empirical data support this solution.

The central premise of our proposed solution, that alleles with fitness benefits to one sex can spread in a population despite costs to the other sex, is supported theoretically here and by earlier models[21]. Our model assumes females are the longer-lived sex. This is true for humans and many primates for which we have high-quality demographic data[22]. We also assume there are alleles that improve male fitness but reduce female health. This assumption is supported by the observation that masculine looking men are both likely to have higher reproductive success and suffer less from some health problems, but masculine looking women have a higher risk of these conditions[18] and by the wealth

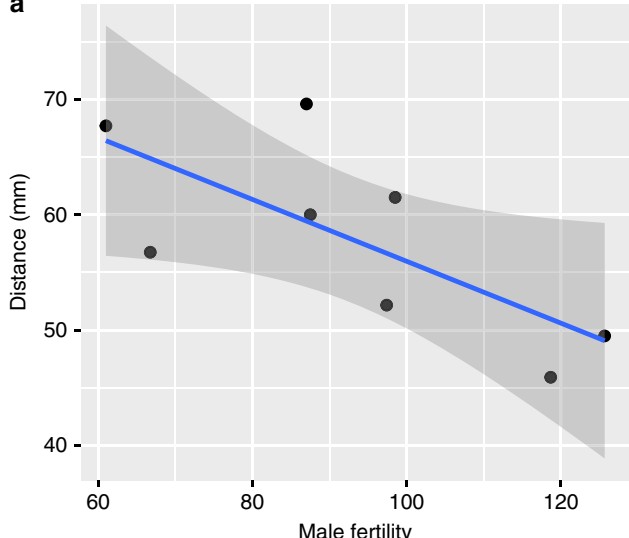

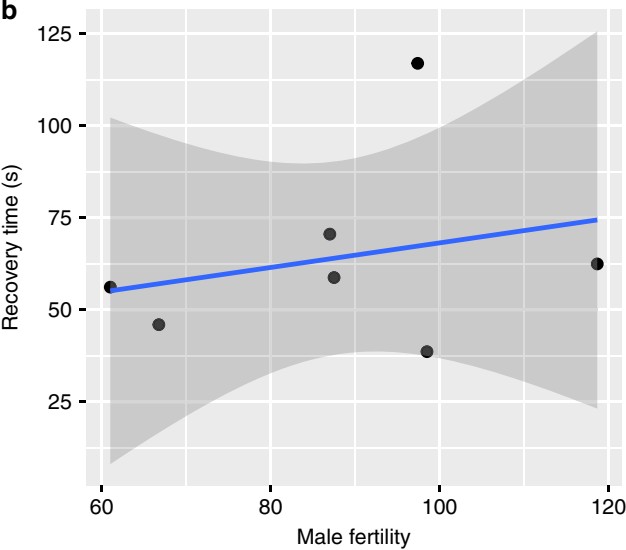

**Fig. 3** Associations between late-life male fitness and female health measures. **a** Male fertility at old age (35 days old) plotted against female performance in negative geotaxis assays (where a high number indicates a long distance travelled and high performance) (correlation coefficient: −0.715, $P$ = 0.046) and **b** female recovery time from anaesthesia (where a large number is a slow recovery and hence a negative indicator of performance) (correlation coefficient: 0.257, $P$ = 0.577) in a sample of iso-female lines (standardised genotypes). Line averages and 95% confidence intervals are shown

of direct evidence showing that negative intersexual fitness correlations are widespread across the animal kingdom[12,23]. While alleles with sexually antagonistic effects are common, their effects could be modified by alleles that alter hormone levels. So for example, sex hormones could affect the expression of shared traits in sex-specific ways, relaxing sexual conflict[24]. However, while sex hormones can relieve sexual conflict, in bank voles there can also be pronounced sexual conflict over optimal levels of circulating sex hormones, and these can lead to negative correlations for fitness across the sexes[25]. Thus, there is the potential for sexual conflict in humans despite a role for sex hormones in generating sexual dimorphism. We also rely on the reductions in health not being so severe that they reduce female lifespan. In other words, we assume that lifespan and health can evolve somewhat

independently in humans. This assumption is supported by the existence of the health-survival paradox. Our idea also relies on there being male-benefit alleles that act after the age of menopause and this relies on men expressing these alleles having reproductive success late-in-life. Although men with higher reproductive success tend to live shorter lives[26], in many societies men can reproduce long after women experience the menopause[19] (even though most male reproduction occurs at ages when women are still reproductively active). In any case, our model shows that as long as males achieve some fitness late-in-life and as long as this does not trade-off with early-life fitness, these alleles will spread through the population.

Additionally, for the conflict hypothesis to be a feasible solution to the health-survival paradox, the male benefits and female fitness costs that promote the spread of sexually antagonistic alleles must be realistic. We assume that the benefits of expressing these alleles range between 1 (i.e. males with and without this allele produce the same numbers of offspring) and 2 (i.e. males with this allele produce twice as many progeny as those without it). In traditional societies, average male reproductive success is around six offspring but in some societies can vary between 0 and 80 children[27]. Therefore, the advantages of expressing male-benefit alleles could be even higher than those we modelled. In females, the costs of expressing late-acting, male-benefit sexually antagonistic alleles are likely to be low. Although impaired health late-in-life could limit how well women provision their grandchildren, and so reduce indirect female fitness[28], these effects will be modest because grandchildren only share around a quarter of their grandmother's genome. Stated another way, direct effects tend be more selectively important than indirect effects. On balance therefore, it seems feasible that late-acting, male-benefit alleles will have greater positive effects on male fitness than costs to females, and so they are likely to spread. However, the female fitness costs of expressing male-benefit, female-detriment alleles have never been fully quantified in humans.

Having shown theoretically that the alleles we envisage could underpin the health-survival paradox, we empirically tested these predictions in a general way using *Drosophila* models. We selected on male fertility late-in-life, and as a correlated response females suffered a reduction in lifespan. This is clearly not a perfect test of the model, if only because our model predicts reduced female health and not survival. Additionally, our control is the founder population from which experimental flies were derived, an approach akin to translocation experiments in ecology and not a "classical" control. Finally, we are testing a model about human health in flies—the nature of the fitness costs of expressing male-benefit alleles will inevitably differ between flies and humans for many reasons, not least because flies do not experience the menopause or receive any indirect benefits from providing parental care. However, it is important to note that the aim of this experiment was simply to see if biasing selection late-in-life towards one sex, can have costly effects on the other. Our data suggest that it can and indeed, this is precisely what an enormous body of evolutionary theory predicts.

We then showed that genotypes with high late-life male fertility produce females that perform less well in negative geotaxis assays, which is a general measure of fly health. Although geotaxis assays are not a perfect health assay, they do suggest that poorly performing females have reduced motor function. This has clear parallels to human data, where elderly women consistently experience greater physical restrictions in their daily lives than men. We did not detect a relationship between male fertility and recovery time from anaesthesia. However, given that human women do not perform worse than men in all individual measures of performance, despite consistently experiencing poorer overall health[29], it is not surprising that we found differences

across our measures of health. A more robust test of our hypothesis would require adopting a similar approach in humans and testing whether men with greater fertility late-in-life have sisters with poor late-life health. If so, this would indicate that females carrying alleles that build a high-quality male suffer more late-in-life. To our knowledge, these data are not available.

In summary, our results indicate that intralocus sexual conflict could have a pivotal role in reducing female health late-in-life and thus provide a general solution to the human male–female, health-survival paradox. To the best of our knowledge, the male–female, health-survival paradox has never been addressed through a conflict lens. However, there are caveats to our experimental evolution data and historical human pedigree data would enable the conflict explanation to be tested more directly. Robust testing of this idea is important, particularly as the age of reproduction in many societies is being pushed later in life. This could have impacts on sex differences in healthy aging if our hypothesis is correct, although the nature of these impacts would depend on whether sex differences in reproductive success late-in-life become relaxed or exaggerated as a consequence of age-related reproductive shifts.

## Methods

**The model**. We used a simple population genetic model to simulate and track the prevalence of a sexually antagonistic allele, $x$, through a diploid population over time. To allow for the allele to have a sex-specific effect and to be either recessive or dominant, we split the populations into males ($m$) and females ($f$) that are further divided according to the allelic composition of the genetic locus in question. That is, we account for individuals that are homozygous wrt $x$ ($m_{xx}$ and $f_{xx}$), homozygous wrt $y$ ($m_{yy}$ and $f_{yy}$) or heterozygous ($m_{xy}$ and $f_{xy}$). We assumed mating to be frequency-dependent and for simplicity assumed the population size remains constant over time. The population dynamics can then be described by the following set of differential equations:

$$\frac{\mathrm{d}m_{ij}}{\mathrm{d}t} = \mu(1-\rho)G_{ij} - \mu_m\,m_{ij}, i,j \in \{x,y\} \tag{1}$$

$$\frac{\mathrm{d}f_{ij}}{\mathrm{d}t} = \mu\,\rho\,G_{ij} - \mu_f f_{ij} \tag{2}$$

with

$$\mu = \sum_{i,j} \mu_m m_{ij} + \mu_f f_{ij} \tag{3}$$

$$G_{xx} = \pi_{xx}^f\left(\pi_{xx}^m + 0.5\pi_{xy}^m\right) + \pi_{xy}^f\left(0.5\pi_{xx}^m + 0.25\pi_{xy}^m\right) \tag{4}$$

$$G_{yy} = \pi_{yy}^f\left(\pi_{yy}^m + 0.5\pi_{xy}^m\right) + \pi_{xy}^f\left(0.5\pi_{yy}^m + 0.25\pi_{xy}^m\right) \tag{5}$$

$$G_{xy} = \pi_{xx}^f\left(\pi_{yy}^m + 0.5\pi_{xy}^m\right) + \pi_{yy}^f\left(\pi_{xx}^m + 0.5\pi_{xy}^m\right) + \pi_{xy}^f\left(0.5\pi_{xx}^m + 0.5\pi_{xy}^m + 0.5\pi_{yy}^m\right) \tag{6}$$

where

$$\pi_{ij}^s = \frac{s_{ij}}{\sum_{i,j} s_{ij}}, s \in \{m,f\} \tag{7}$$

is the proportion of males or females with locus ($ij$). $\rho$ denotes the female sex bias (if considered). We further assumed that the cost or benefit associated with carrying allele $x$ is solely manifested through a decrease or increase in lifetime reproductive success. We cannot model costs to females as reduced health per se; if reductions in health do not reduce female fitness then trivially, any male-benefit alleles will spread. We therefore assume that alleles that severely reduce female health, reduce female fitness by reducing how well women care for their offspring and grand-offspring. Under this assumption, a shortening or lengthening of an individual's reproductive period is equivalent to a reduction or increase in reproductive fitness. In the model this can be incorporated by changing $\pi_{ij}^s$ as follows

$$\pi_{ij}^m = \frac{b_{ij} m_{ij}}{\sum_{i,j} b_{ij} m_{ij}} \tag{8}$$

and

$$\pi_{ij}^f = \frac{c_{ij} m_{ij}}{\sum_{i,j} c_{ij} f_{ij}} \tag{9}$$

where $b_{ij} \geq 1$ denotes the male-benefit and $c_{ij} \leq 1$ denotes the cost to females. Considering $x$ to take an effect only in homozygous form this results in

$$b_{xx} \neq 0, b_{xy} = 0, b_{yy} = 0$$

$$c_{xx} \neq 0, c_{xy} = 0, c_{yy} = 0$$

whereas the situation in which $x$ is dominant can be described as

$$b_{xx} \neq 0, b_{xy} \neq 0, b_{yy} = 0$$

$$c_{xx} \neq 0, c_{xy} \neq 0, c_{yy} = 0$$

This formulation also allows us to consider dominant and recessive effects in females and males differentially. To obtain equilibrium population frequencies, we solved Eqs. (1) and (2) numerically using the odeint solver from the NumPy Python package. For each combination of male-fitness benefits ($b_{ij}$) and female fitness costs ($c_{ij}$), we simulated the model forward in time until an equilibrium point was reached, using the following initial conditions: $m_{xx}(0) = 0.05$, $m_{xy}(0) = 0.3$, $m_{yy}(0) = 0.1$, $f_{xx}(0) = 0.1$, $f_{xy}(0) = 0.35$, $f_{yy}(0) = 0.1$.

**Selecting on late-life male fertility affects female lifespan.** If intralocus sexual conflict is responsible for the health-survival paradox, then selecting for late-life male reproduction should result in the accumulation of late-acting male-benefit alleles that reduce female fitness. To broadly test this idea, we selected for late-life male fertility in replicate populations of *Drosophila simulans* but relaxed selection on females. If sexual conflict is responsible for the health-survival paradox, then in populations where males experienced the greatest increase in fertility late-in-life, females should experience the greatest reduction in lifespan (relative to controls).

Selecting on male fertility involved establishing five experimental populations and five female-supply populations (see below) using flies collected from our large outbreeding, free-mating lab-stock population (100 males and 100 females (all virgins) per experimental and female-supply population). For 28 days, fly food in the experimental populations was changed every five days to ensure that no eggs laid emerged as adults. On day 28, flies were anaesthetised with $CO_2$, and removed from the cage. One hundred virgin 3- to 5-day-old females were taken from their paired-female-supply populations (to ensure experimental populations were independently evolving), and added to the appropriate experimental population. This should reduce selection for old age reproduction in females. Old (28 days) males from each experimental population and young females from paired-female supply were then left for three days to lay eggs. From these, 100 male and 100 female offspring were collected on emergence and these seeded the next generation in each experimental population. This procedure was repeated for 12 generations. Female-supply populations were also fed every five days, but here new food was added on day 15, removed on day 18 and offspring collected from eggs laid between days 15 and 18. This regime is shown in Supplementary Figure 1.

To test for male responses to our artificial selection, we collected 30 virgin males from each experimental population and from the lab-stock population. Each male was housed in a 40 mL vial, with 8 mL of medium for two days to mature. Then two virgin lab-stock females were added and these were replaced with young virgin females every week to ensure males had continuous access to young virgins (as is the case in selection cages). Any female that died was replaced with a like-age virgin. On day 28, male fertility was assayed. Each focal male was paired with a virgin stock female. Males were removed eight hours later. Females were allowed to lay eggs in three vials over seven days and all offspring that emerged from these vials were counted as our measure of male fertility (full details in Supplementary Figure 2). Males that did not appear to mate (i.e. counts of offspring = zero), were excluded from analyses.

To test for female impacts resulting from selecting on male fertility, 40 females from each experimental population and the lab-stock population were collected as virgins and housed individually in 40 mL vials contain 8 mL of Jazz mix medium. After four days, flies were moved into a new vial. The following morning two stock virgin males (three to five days old) were added and kept with the females for three hours before being removed. This regime was repeated across the entire lifespan of each female fly (full details in Supplementary Figure 3). Pairing females with two males for three hours every five days ensures that females reproduce as normal but lifespan is not reduced due to the direct costs of mating or harassment[30]. Females were checked daily for survival and adult lifespan was calculated.

To assess how late-life male fertility evolved in the experimental populations, mean male fertility in the stock population was subtracted from the mean value for each experimental population. High positive values meant that males from the experimental population performed much better than non-selected males, and a negative value meant that experimental males had worse late-life fertility. Female

lifespan was treated in the same way (i.e. average population value—average stock-population value) and a Pearson's correlation coefficient was calculated in R version 3.4.1.[31] to analyse the correlation between male late-life fertility and female lifespan.

**Association between late-life male fitness and female health.** The above "selection then assay protocol" tests the general principle of the health-survival paradox, but does not directly test for female health declines as late-life male fitness increases. If intralocus sexual conflict is responsible for the health-survival paradox, then genotypes that produce high fitness males should produce females with poor health and vice versa. This was tested using *Drosophila melanogaster* Genetic Reference Panel (DGRP) lines and two biomarkers of female physical function to reveal underlying health.

Ten DGRP lines were used (ID = 28, 101, 136, 360, 382, 443, 595, 737, 783, 796). Flies were maintained under a 12:12 L:D cycle at 25 °C. Lines were excluded from analyses, if fewer than two animals survived to assay. The final sample sizes for each line and trait are given in Supplementary Table 1. Experimental flies were collected as virgins and aspirated into individual 100 mL vials on their day of hatching with 8 mL of medium. Dahomey tester flies were used as mating partners to assess reproductive performance. Ivanov et al.[32] recorded lifespan in DGRP lines and median lifespan ranged between 21 and 79 days. We therefore conducted tests when adults were 35 days old, approximately the median lifespan for some of the shorter lived genotypes.

Negative geotaxis (vertical climbing in response to shock) was one measure of fly health. It is a measure of motor ability that shows an age-dependent decline in *Drosophila*[33]. To assay negative geotaxis, flies were aspirated into 15 mL vials attached to a drop mechanism, which was raised 10 cm and dropped. A camera recorded every trial, to record the distance that flies climbed in the two minutes after dropping. Flies were then given two minutes to recover before the process was repeated. All observations were made blind—flies were labelled with a random number and videos were analysed independently by two different observers, and any values that differed by >3 mm were observed by a third experimenter to reach consensus. Recovery time from anaesthesia was also used as a measure of female health, as this can indicate metabolic performance. Assay flies were transferred into a 15 mL vials where they were exposed to $CO_2$ (1 L/min) for thirty seconds. Flies were then put onto a piece of white paper and the time until flies stood upright was recorded and used as our measure of performance. All assays were recorded within two hours of lights going (i.e. 9–11 am) and flies assayed in a random order. Measures were made blind; flies were labelled with a random number by one lab member, before vials were passed to the observer.

Reproductive performance of the DGRP males was assessed after matings with virgin females from our wildtype stock animals. Each experimental male was housed with two virgin tester females, aged between 3 and 6 days old, that had been left overnight in 40 mL mating vials containing surplus food. Flies were then left for 48 h, after which, females were transferred to a vial for oviposition for a further 48 h, while males were removed and frozen. Females were then moved to one more vial for a further 48 h, such that their egg laying over 6 days was recorded. Oviposition vials were then incubated at the temperature from which their sire originated and offspring were counted 8 days after the first day of offspring eclosion.

Please note, differences in timings between the two male fertility assays represent species-specific lab protocols. Additionally, males were kept as virgins prior to being assayed in this experiment to allow comparison with females who were also maintained as virgins to ensure that direct physiological damage caused by male harassment did not reduce their physical performance.

To test for effects, we created a single average value, for each line and each trait for analyses (i.e. we conducted derived variable analyses). For male fertility, zero counts were once more excluded as we could not be sure that males had mated if no offspring were produced, but note this is conservative for our hypothesis. For female geotaxis distance, we included zero values but note that the results of analyses are the same irrespective of whether we include all data for both traits or exclude zeros. We then used a Pearson's correlations, calculated in R, to analyse the associations between male late-life fertility, and female health measures.

## Data availability

The data that support the findings of this study are archived in the Dryad Digital Repository (https://doi.org/10.5061/dryad.p888tv2). A Reporting Summary for this article is available as a Supplementary Information file.

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

## Acknowledgements
Thanks to all members of DJH's lab group for useful discussions that informed this manuscript and help collecting data or maintaining fly stocks: Cleo Alper, Mathilda Janicot Bale, Matt Carey, Jacqui Glencross, Noam Hosken, Tomo Noda, Sam Oyesiku-Blakemore, MD Sharma, Stefan Store, and Andreas Sutter. We thank the National Science Center (Poland: 2013/09/N/NZ/NZ8/03231) and the Leverhulme Trust (UK: RF-2015-01) for funding, which partially supported this work, and the University of Exeter's Dean's Fellowship for additional support.

## Author contributions
C.R.A. and D.J.H. conceived the study and wrote the manuscript, M.R. devised the mathematical model, C.R.A., D.J.H., and E.D. designed the experiments, C.R.A. and E.D. collected data, all authors helped edit the manuscript.

## Additional information

**Competing interests:** The authors declare no competing interests.

