## [Peer Review File · Nature Communications]

Reviewers' Comments:

Reviewer #1:

Remarks to the Author:

Review for Archer, Recker, Duffy & Hosken: Why do men die, while women suffer? Intralocus sexual conflict resolves the male-female, health-survival paradox.

This is a fascinating paper that attempts to explain the male-female, health- survival paradox which is so far only described in humans, by ways of a novel approach (to me at least), namely intralocus sexual conflict. The authors use three approaches to make their point: A modelling approach, a selection experiment, and a comparative study using several strains of *Drosophila*. The authors convincingly show that the mechanism they describe can generate mortality and health patterns comparable to the health survival paradox in humans.

The paper is well written, and very understandable. The authors do an excellent job in explaining their theory, and why it is a plausible candidate in explaining the health-survival paradox in humans. Also, I could not find a flaw in their methodology, and analyses. I did not understand the intricacies of the model too well, but the results of the model are well presented and understandable.

I believe the paper has a broad audience, and is definitely of general interest to the general readership, and it does a great job describing the effect of intralocus sexual conflict at old age.

Whether this mechanism is actually responsible for the human phenomenon remains to be seen, and this reviewer's only caveat is that selection on male traits at an age later than females at menopause may be quite weak, as the contribution to fitness at these old ages may be negligible, as only few old males sire offspring at ages older than 50 or 60. There is only one Charlie Chaplin in a billion, and the high contribution of old males in Cameroon 1964 in the Tuljapurkar et al. paper may be an exception.

Recommendation: accept; no minor revision

Alex Scheuerlein

Reviewer #2:

Remarks to the Author:

This is an intriguing and highly readable paper concerning a paradox in human ageing, that men tend to die younger whereas women suffer worse health. The authors present a simple population genetic model of the idea that sex-dependent selection on late-life male reproduction, beyond the menopause, might be responsible for women's worse late-life health via the accumulation of late-acting male-benefit sexually antagonistic alleles.

I like the verbal articulation of the idea and the model, but I am not a modeller able to assess this part in detail.

The authors then present the result of five lines of independently evolving flies, where late-life male but not female reproduction was selected for via the husbandry method. They show some evidence (though not strong effects) that this regime indeed results in weaker general female health.

My major concern with this manuscript is that this fly study is not an experiment. The comparison was with the group's stock fly populations. This the 'treatment' applied far more changes, relative to the comparison, than those of interest. This is an elementary design flaw, and I am surprised to see it in a paper of such interesting conceptual quality. I know that designing these kinds of ecological selection

studies is difficult, but that is why a number of procedural controls are often put in place. My opinion is that such a big flaw in design will do the journal and the authors no credit, if published, and I encourage the authors to replicate their result with replicate sets of one or more appropriate controls.

The question of sexually antagonistic genes being involved in ageing differences, though interesting and as yet unresolved, is not the terra incognita this rather overconfident paper makes it out to be. In the past 10-12 years there have been suggestions that a mechanism like this may be at play in humans, other mammals and insect models. So I don't think it is feasible to use the rather loose *D.simulans* studies as a sketchy illustration of a new and iconoclastic point: the idea requires far more rigorous scrutiny.

Minor point - Lines 61-63: The Chippindale-Rice example of hip width is a speculation, now largely refuted, and should not be used as evidence of sexually antagonistic selection.

Reviewer #3:

Remarks to the Author:

This is a fascinating and original study that sheds light on a centrally-important question: why are there differences in male and female ageing and health in humans? Using a multi-disciplinary combination of theoretical and experimental evolution approaches, the work shows that intralocus conflict could provide key insights and explanations to human health and ageing.

Intralocus conflict occurs because there are two different sexes requiring different functions to operate optimally, but both sexes must share a common genome. Good alleles for male function can therefore be detrimental to female function, and vice versa. ILC could be especially important in humans because of our clear dimorphism in age-specific fertility and reproduction. Men can and do reproduce at much later ages than women, who are constrained in their age-based reproductive output by a biological cliff-edge effect from menopause. Indeed, this old-age-dimorphism in reproduction may be under increasing pressure in some societies where reproduction is taking place later in life, and the authors might consider adding this consideration into the manuscript, or thinking about testing different cultures that vary in this regard?

The model is sensibly constructed and importantly demonstrates that ILC can operate to impact on female health as a consequence of male late-age reproduction across a number of conditions. It was good to see the incorporation of inclusive fitness benefits from relevant 'grandmother helping' effects in the theory.

The two sets of experiments provide compelling evidence in support of ILC between male late-age reproduction and female fitness. Of course there are differences between humans and flies, but the powerful tool of experimental evolution has been used here to complement the model and the fit to a vast knowledge on human ageing and health. In lines selected for improved late-age male reproduction, there was evidence of more detrimental impacts on female function, supporting the thesis of ILC here.

In general, I see this as a novel, exciting, and stimulating piece of work that importantly applies both evolutionary theory and experimental approaches to tackling enormous (and growing) questions in human health.

I have two suggestions concerning background and interpretation for the authors to consider. The first concerns ILC modulation and hormonal control, which I think would be a first thought for biomedical

researchers. Nature Comms is a general journal with a broad target audience, especially if the work can be applied to biomedical researchers. Although the authors do a generally great job of avoiding evolutionary jargon and explain ILC well, it may help to explain further how ILC can exist, despite the important modulation beyond the genome that can occur via hormonal control. I can imagine a lot of medics thinking that ILC is bogus for humans (and other mammals) because hormones modulate any genomic conflicts, and it is the sex hormones that allow males to become more male, and females to become more female. Some brief discussion of how ILC interacts with hormonal attenuation, and of relevance to humans, would help.

Second, it would help to clarify the basis of the human male-female, health-survival paradox with regard to male survival as a consequence of male risk-taking behaviour and lifestyle. Although not necessarily key to the theory, it would be worthwhile adding in some discussion / consideration about genes versus environment within human (male) elevated mortality.

RESPONSE TO REVIEWERS

REVIEWER #1

General Feedback: This is a fascinating paper that attempts to explain the male-female, health- survival paradox which is so far only described in humans, by ways of a novel approach (to me at least), namely intralocus sexual conflict. The authors use three approaches to make their point: A modelling approach, a selection experiment, and a comparative study using several strains of *Drosophila*. The authors convincingly show that the mechanism they describe can generate mortality and health patterns comparable to the health survival paradox in humans.

The paper is well written, and very understandable. The authors do an excellent job in explaining their theory, and why it is a plausible candidate in explaining the health-survival paradox in humans. Also, I could not find a flaw in their methodology, and analyses. I did not understand the intricacies of the model too well, but the results of the model are well presented and understandable.

I believe the paper has a broad audience, and is definitely of general interest to the general readership, and it does a great job describing the effect of intralocus sexual conflict at old age. Whether this mechanism is actually responsible for the human phenomenon remains to be seen, and this reviewer's only caveat is that selection on male traits at an age later than females at menopause may be quite weak, as the contribution to fitness at these old ages may be negligible, as only few old males sire offspring at ages older than 50 or 60. There is only one Charlie Chaplin in a billion, and the high contribution of old males in Cameroon 1964 in the Tuljapurkar et al. paper may be an exception.

Recommendation: accept; no minor revision

We were really pleased that you found the manuscript fascinating, convincing and well written and thank the Reviewer for the feedback. The Reviewer is quite right that while men can reproduce until advanced ages, this is certainly not the norm in all cultures, although we would still argue that male reproduction occurs at reasonable frequency after females enter menopause. We now highlight that while men are able to reproduce at late ages, this may not be a particularly common event and say, "*Although men with higher reproductive success tend to live shorter lives²⁶, in many societies men can reproduce long after women experience the menopause¹⁹ (even though most male reproduction occurs at ages when women are still reproductively active)*" (Page 6, Line 187 – 190)."

REVIEWER #2.

General Feedback: This is an intriguing and highly readable paper concerning a paradox in human ageing, that men tend to die younger whereas women suffer worse health. The authors present a simple population genetic model of the idea that sex-dependent selection on late-life male reproduction, beyond the menopause, might be responsible for women's worse late-life health via the accumulation of late-acting male-benefit sexually antagonistic alleles. I like the verbal articulation of the idea and the model, but I am not a modeller able to assess this part in detail.

The authors then present the result of five lines of independently evolving flies, where late-life male but not female reproduction was selected for via the husbandry method.

They show some evidence (though not strong effects) that this regime indeed results in weaker general female health.

We were really pleased that the Reviewer finds the paper readable and intriguing. We thank them for their feedback and we hope we have been able to address any concerns adequately.

Reviewer 2, Comment 1. My major concern with this manuscript is that this fly study is not an experiment. The comparison was with the group's stock fly populations. This the 'treatment' applied far more changes, relative to the comparison, than those of interest. This is an elementary design flaw, and I am surprised to see it in a paper of such interesting conceptual quality. I know that designing these kinds of ecological selection studies is difficult, but that is why a number of procedural controls are often put in place. My opinion is that such a big flaw in design will do the journal and the authors no credit, if published, and I encourage the authors to replicate their result with replicate sets of one or more appropriate controls.

First, we would like to note that there were two fly data-sets used, but only the experimental evolution data seemed to cause any concern. We completely agree that there was not a perfect control for this experimental evolution element of the study. While agreeing that the use of the founding population as a control is not perfect, we also note there is no way to generate what might typically be considered the classical control. It is simply not possible to create control populations that have long lives without selecting on late-life male fertility, i.e. all else is the same but there is no selection on male fertility, if we wanted to keep the generation times in synch. It could not be done without inadvertently selecting on late-life reproduction, which is our experimental treatment.

This is because to generate a classic control would require following the same schedule in control populations as the experimental populations (e.g. population turn-over at day 28) – otherwise they are not really acting as controls - and both males and females would have had to reproduce late-in-life to contribute to the next generation. This would not make an adequate control because flies would experience selection for late-life reproduction, which is our experimental treatment. Furthermore, if there were any sexually antagonistic alleles for late-life reproductive success, as there appear to be (and indeed, this is what we are testing for), and sexual selection is stronger on males than females (which it is), we would also generate negative correlations between male and female late-life fertility in the controls. As a result, controls would mirror the experimental treatment in all ways. In fact we initially established populations like these as controls until we realised that all we were doing was replicating the “experimental” populations with a smaller effective population size. Alternatively, “controls” would reproduce at young age so as to not select on late-life male fertility, and hence “controls” would have to go through many more generations of evolution, and that means comparing apples with pears, which is no control, and which is why we could not do this. As a result, we took the pragmatic decision to use the founding population – *the population that provided the genes and phenotypes for all our experimental populations* - as the comparator. And while not perfect, if we make a specific prediction that is met, the resulting data are not without merit and provide some interesting information we think.

Accordingly, although we were unable to provide classical controls, we were able to make specific predictions about how the experimental populations should differ from the founder population based on our model prediction, and we found exactly the pattern we would expect if predictions were correct. That is, there should be negative associations between male and female traits, and critically, this relationship depends on how well our artificial selection on

males worked. In other words, how far males in each population increased from the founder values should predict how far females deviated negatively relative to the founder, with the size of the negative female deviation in each population depending on the size of the male increase. This is pretty specific and exactly what we saw.

To be clear, the prediction was that the more populations get better at late-life male reproduction relative to base-line, the worse that females from those populations should become relative to base-line. Although we do not have classical controls we do have a base-line: the founding population from which all our experimental populations are derived. Furthermore, using the starting population as a control group *is common practice* in experimental evolution and other studies. For example, Allen et al. (*BMC Evolutionary Biology* [2008] 8:94) present responses to selection as phenotypic standard deviations from the starting population mean, and Tobler et al. (*Molecular Biology and Evolution* [2013] 31:364-375) compared gene expression in populations evolved to hot or cold environments to the founder population. To us, those and our approach is similar to translocation experiments in ecology where morphology in a founder population is compared to island translocations (e.g. anolis lizard work: for an overview see *Anolis Lizards of the Caribbean: Roughgarden [1995]* and work by Losos <https://lososlab.oeb.harvard.edu/publications?page=1>), and these studies have been incredibly revealing and not seen as flawed. Similarly, we do not agree with characterising this element of our study as “flawed”.

In summary, we agree there are confounders, but nonetheless, to generate the exact pattern predicted by our model is at least interesting and broadly supportive of the explanation we propose for the health-survival paradox in humans. We also note that in the original submission we were extremely clear that flies are not humans and so our data are only indicative of the possibility that this idea could explain a human condition.

We have tried to be much more explicit about all of these caveats, both in the main body of the text and in the figure legends, and hope that we have done so in a manner that is broadly acceptable. If it would help to ensure readers were aware of the caveats, we could move the figure to supplementary materials. But to reiterate, the fly data – both data-sets – provide evidence that is at least partly consistent with the model prediction. While flies clearly differ from humans in innumerable ways, these data match model prediction and show that at least in principle, our verbal and mathematical arguments could have some biological relevance. We hope that on reflection you agree. We would of course be open to any further suggestions about how we might additionally address your concerns.

We have inserted the following into the main body of the text:

Page 7, Line 213 – 223: “This is clearly not a perfect test of the model, if only because our model predicts reduced female health and not survival. Additionally, our control is the founder population from which experimental flies were derived, an approach akin to translocation experiments in ecology and not a “classical” control. Finally, we are testing a model about human health in flies - the nature of the fitness costs of expressing male-benefit alleles will inevitably differ between flies and humans for many reasons, not least because flies do not experience the menopause or receive any indirect benefits from providing parental care. However, it is important to note that the aim of this experiment was simply to see if biasing selection late in life towards one sex, can have costly effects on the other. Our data suggest that it can and indeed, this is precisely what an enormous body of evolutionary theory predicts.”

And:

Page 8, Line 240 - 242: "However, there are caveats to our experimental evolution data and historical human pedigree data would enable the conflict explanation to be tested more directly..."

And in the figure legend we once more highlight the nature of our control population to be entirely explicit about this:

Page 19, Line 509 – 513: "Changes in female lifespan were regressed against improvements in male fertility in populations subject to artificial selection for male late-life fertility relative to the stock population from which experimental flies were derived (correlation coefficient = -0.93; $P = 0.022$). Thus the stock population acts as a base-line against which evolution was assessed."

Reviewer 2, Comment 2. The question of sexually antagonistic genes being involved in ageing differences, though interesting and as yet unresolved, is not the terra incognita this rather overconfident paper makes it out to be. In the past 10-12 years there have been suggestions that a mechanism like this may be at play in humans, other mammals and insect models. So I don't think it is feasible to use the rather loose *D.simulans* studies as a sketchy illustration of a new and iconoclastic point: the idea requires far more rigorous scrutiny.

Again we are in total agreement about conflict and aging, as this is an area where our research has made valuable contributions, including illustrating that intralocus sexual conflict has many consequences across species and across the genome (see e.g. *Current Biology* [2010] 20:2036-2039; *Ecology Letters* [2012] 15:193-197; *BMC Biology* [2015] 13:34; *Proceedings B* [2016] 283:20161429; *Evolution* [2011] 65:2133-2144) and that intralocus sexual conflict affects sex-specific patterns of aging (e.g. *Behavioural Ecology* (in press); *Functional Ecology* [2015] 29:562-569; *Evolution* [2012] 66:3088-3100; *Evolution* [2013] 67:620-634; *Experimental Gerontology* [2015] 71:4-13). We also agree that there is a long history of sexual conflict being identified as a driver of sex differences in aging and lifespan and a wealth of empirical evidence supporting a role for sexual conflict in actuarial aging. There is also a body of data showing that sexual conflict might be implicated in driving sex-differences in human health. However, what there has not been, to our knowledge, is any reference of sexual conflict in relation to the health-survival paradox. Neither have we seen any suggestions that sexual conflict is more likely to drive sex-differences in health after the age of the menopause, which is the fundamental point of this paper.

To be clear, we were not trying to claim that a role for sexual conflict in aging is new, but we are suggesting that sexual conflict could play a role in the male-female, health-survival paradox and that so far, this possible role has been overlooked. We had hoped that we were clear about this in the initial version of our manuscript where we said:

"While we recognise that intralocus conflict can cause sex differences in aging and health¹², its potential contribution to the human health-survival paradox has been overlooked."

So we were aware of all this and were a little puzzled that this point was made in review because we thought we were very clear that we were making a very specific but important point: we were testing the human health-survival paradox which has never been addressed through a conflict lens. Although if we have somehow overlooked solutions to the paradox based on conflict it would be very helpful to have guidance to that literature. We have revised the manuscript to be clearer (we hope) about what we are addressing and we trust this is satisfactory. We apologise for appearing "over confident" and again cordially request

that if we have overlooked conflict research into the health-survival paradox, any guidance to references would be excellent. In the interim, we have written the following:

Page 3, Line 72-81: "A role for sexual conflict in aging and lifespan and sex differences in health has been recognised for years^{13,14}. However, to the best of our knowledge, sexual conflict has not been considered as a driver of the male-female, health-survival paradox. Given the existence of the menopause, which enables selection to bias allelic values towards male-benefit late-in-life, there is enormous potential for sexual conflict to be at the heart of this paradox and by recognising its role, we may better understand what (if anything) we can do about it. The aim of this paper is simply to highlight that as well as explaining sex differences in health and aging in general, intralocus sexual conflict could be central to a long-standing puzzle in medical sciences: why do men die, while women suffer?"

We further highlight the specific topic of our study in the final paragraph of the discussion, where we reiterate, "*To the best of our knowledge, the male-female, health-survival paradox has never been addressed through a conflict lens*" (Page 8, 239-240).

As a point of clarification, you refer here to *D. simulans* studies (plural), which we assume means just the experimental evolution work as we used two species of *Drosophila* here to have a broader number of model systems to interrogate the theory (i.e. any support was not due to the idiosyncrasies of one species). We additionally note that insects, including *Drosophila*, are routinely used as models for human research, from cancer to Alzheimer's to arthritis, and so we do not see that using other biological systems to make points about human biology is "sketchy". Further, we also think using mathematical models provides a reasonably "rigorous" form of scrutiny, especially when coupled with experiments in two different species that provide biological support for the theory. We trust that this is now OK but again are happy to further revise if there are outstanding concerns.

Reviewer 2, Comment 3. Lines 61-63: The Chippindale-Rice example of hip width is a speculation, now largely refuted, and should not be used as evidence of sexually antagonistic selection.

We are not 100% sure this is refuted given that the sex of a human skeleton can be determined by hip-design. That is, male and female hips in humans differ, which means there had to be sex-specific selection for hip-width and if hip-width had (or has) a common genetic basis, then by definition there is (was) conflict over optimal hip values. However, we have replaced this example with work on beetles which shows that when females inherit alleles that promote male fitness (in this case large mandibles that males use to fight), they have reduced fitness even though females do not express this trait. This section now reads:

Page 3, Line 66-71: "For example, male broad-horned flour beetles develop enlarged mandibles and males with larger mandibles have higher fitness. However, daughters of males with large mandibles have lower fitness because of the masculinisation of the body that occurs with these genotypes¹². This means that alleles associated with mandibles are subjected to an intersexual tug-of-war over optimal values, with high fitness male genotypes making low fitness females."

Again thank you for the time you put into the review plus the comments, which have helped us really sharpen the manuscript, and we trust we have allayed your concerns.

REVIEWER #3

General Feedback: This is a fascinating and original study that sheds light on a centrally-important question: why are there differences in male and female ageing and health in humans? Using a multi-disciplinary combination of theoretical and experimental evolution approaches, the work shows that intralocus conflict could provide key insights and explanations to human health and ageing.

We were very pleased that the Reviewer found this a fascinating and original study. We also thank the Reviewer for their feedback and suggestions.

Intralocus conflict occurs because there are two different sexes requiring different functions to operate optimally, but both sexes must share a common genome. Good alleles for male function can therefore be detrimental to female function, and vice versa. ILC could be especially important in humans because of our clear dimorphism in age-specific fertility and reproduction. Men can and do reproduce at much later ages than women, who are constrained in their age-based reproductive output by a biological cliff-edge effect from menopause. Indeed, this old-age-dimorphism in reproduction may be under increasing pressure in some societies where reproduction is taking place later in life, and the authors might consider adding this consideration into the manuscript, or thinking about testing different cultures that vary in this regard?

Thanks for this and an interesting point. We have now referenced this possibility in the last line of the Discussion and say, "*Robust testing of this idea using human data is important, particularly as the age of reproduction in many societies is being pushed later in life. This could have impacts on sex differences in healthy aging if our hypothesis is correct, although the nature of these impacts would depend on whether sex differences in reproductive success late-in-life become relaxed or exaggerated as a consequence of age-related reproductive shifts.*" (Page 8, line 242 - 247).

The model is sensibly constructed and importantly demonstrates that ILC can operate to impact on female health as a consequence of male late-age reproduction across a number of conditions. It was good to see the incorporation of inclusive fitness benefits from relevant 'grandmother helping' effects in the theory.

The two sets of experiments provide compelling evidence in support of ILC between male late-age reproduction and female fitness. Of course there are differences between humans and flies, but the powerful tool of experimental evolution has been used here to complement the model and the fit to a vast knowledge on human ageing and health. In lines selected for improved late-age male reproduction, there was evidence of more detrimental impacts on female function, supporting the thesis of ILC here.

In general, I see this as a novel, exciting, and stimulating piece of work that importantly applies both evolutionary theory and experimental approaches to tackling enormous (and growing) questions in human health.

Reviewer 3, Comment 1. I have two suggestions concerning background and interpretation for the authors to consider. The first concerns ILC modulation and hormonal control, which I think would be a first thought for biomedical researchers. Nature Comms is a general journal with a broad target audience, especially if the work can be applied to biomedical researchers. Although the authors do a generally great job of avoiding evolutionary jargon and explain ILC well, it may help to explain further how ILC can exist, despite the important modulation beyond the genome that can occur via hormonal control. I can imagine a lot of medics thinking that ILC is bogus for humans (and other mammals) because hormones modulate any genomic conflicts, and it is the sex hormones that allow males to become more male, and females to

become more female. Some brief discussion of how ILC interacts with hormonal attenuation, and of relevance to humans, would help.

We thank the Reviewer for highlighting that this edit would make the manuscript more appealing to a wider audience and have added some discussion of this to the manuscript. We have added this point to the revised manuscript.

Page 6, Line 176 - 182: "While alleles with sexually antagonistic effects are common, their effects could be modified by alleles that alter hormone levels. So for example, sex-hormones could affect the expression of shared traits in sex-specific ways, relaxing sexual conflict²⁴. However, while sex-hormones can relieve sexual conflict, in bank voles there can also be pronounced sexual conflict over optimal levels of circulating sex hormones, and these can lead to negative correlations for fitness across the sexes²⁵. Thus there is the potential for sexual conflict in humans despite a role for sex-hormones in generating sexual dimorphism."

Reviewer 3, Comment 2. Second, it would help to clarify the basis of the human male-female, health-survival paradox with regard to male survival as a consequence of male risk-taking behaviour and lifestyle. Although not necessarily key to the theory, it would be worthwhile adding in some discussion / consideration about genes versus environment within human (male) elevated mortality.

The Reviewer is right that behavioural differences certainly have the potential to contribute to the paradox. We have now added some discussion of this to the manuscript.

Page 3, Line 52 - 60: "Social and behavioural differences might contribute to these sex differences in survival and health. For example, men are often considered to be more likely to engage in risky behaviours (e.g. smoking) than women, but sex differences in mortality are present even in communities where both sexes avoid risky behaviours⁹. Similarly, men and women with similar chronic medical conditions are equally likely to report health problems¹⁰. While sex differences in behaviour may contribute towards the male-female, health-survival paradox, the consistent observation of lower female mortality but poorer female health at older ages in human populations across the world suggests that the health-survival paradox has, at least in part, a genetic basis."

Reviewers' Comments:

Reviewer #2:

Remarks to the Author:

The most problematic part of the manuscript, for me, was the lack of a suitable control in the *Drosophila simulans* studies. As far as I am concerned the rest of the manuscript was publishable with minor revisions, and I appreciate the way the authors have altered their rhetoric to give greater balance.

The authors have deployed a number of arguments as to why they still think their *D simulans* results stand as a useful test. I am not convinced by these arguments; yes other studies have used the source population as a reference point, but the best among those have employed selection in two directions and this pseudo-control is simply a reference point. There are many other studies that have dealt with the problem of different generation times whilst studying longevity, going back to the classic work by Michael Rose, as well as Rice and Chippindale's contributions to the genetic conflict literature. A single control might not do the job, but a series of well-designed controls are certainly conceivable and I am sure the authors know this.

On balance, I do believe that the *simulans* study does tell us something, and the focus on differences among lines suggests the interpretation favoured by the authors may indeed be true.

I note that the authors themselves acknowledge that this component is not a test of the idea that selection on late life reproduction in one sex can negatively impact health in the other, but rather that it can cause reduced longevity in the other. So this component is much closer to the many other studies of antagonistic gene effects.

On balance, I leave the decision on the quality of evidence required for publication in the editor's hands. Personally, I think a cool proof-of-concept study like this can still do a great service to science, even if one of the lines of evidence falls below the normal standards of evidence. The *simulans* study could not be published stand-alone, but as confirmatory evidence it does a useful job.

The changes in emphasis in this version of the manuscript make the work as a whole less vulnerable to being dismissed on the basis of this problem with experimental design.

Reviewer #3:

Remarks to the Author:

I thank the authors for their attention to my comments about further interpretation and explanation. I also note the sensible responses to the other two reviews and that these further strengthen the work. This is now a really excellent manuscript that both advances our understanding of, and attention to, a major issue within human health and ageing.

REVIEWERS' COMMENTS:

Reviewer #2 (Remarks to the Author):

The most problematic part of the manuscript, for me, was the lack of a suitable control in the *Drosophila simulans* studies. As far as I am concerned the rest of the manuscript was publishable with minor revisions, and I appreciate the way the authors have altered their rhetoric to give greater balance.

Thanks for this, we tried to be more balanced and appreciate that you can see this too.

The authors have deployed a number of arguments as to why they still think their *D simulans* results stand as a useful test. I am not convinced by these arguments; yes other studies have used the source population as a reference point, but the best among those have employed selection in two directions and this pseudo-control is simply a reference point. There are many other studies that have dealt with the problem of different generation times whilst studying longevity, going back to the classic work by Michael Rose, as well as Rice and Chippindale's contributions to the genetic conflict literature. A single control might not do the job, but a series of well-designed controls are certainly conceivable and I am sure the authors know this.

On balance, I do believe that the *simulans* study does tell us something, and the focus on differences among lines suggests the interpretation favoured by the authors may indeed be true.

Thank you.

I note that the authors themselves acknowledge that this component is not a test of the idea that selection on late life reproduction in one sex can negatively impact health in the other, but rather that it can cause reduced longevity in the other. So this component is much closer to the many other studies of antagonistic gene effects.

On balance, I leave the decision on the quality of evidence required for publication in the editor's hands. Personally, I think a cool proof-of-concept study like this can still do a great service to science, even if one of the lines of evidence falls below the normal standards of evidence. The *simulans* study could not be published stand-alone, but as confirmatory evidence it does a useful job.

Again thanks, really appreciate the support.

The changes in emphasis in this version of the manuscript make the work as a whole less vulnerable to being dismissed on the basis of this problem with experimental design.

Reviewer #3 (Remarks to the Author):

I thank the authors for their attention to my comments about further interpretation and explanation. I also note the sensible responses to the other two reviews and that these further strengthen the work.

This is now a really excellent manuscript that both advances our understanding of, and attention to, a major issue within human health and ageing.

Thank you.